# Antidistillation Sampling

**Yash Savani**[*]  **Asher Trockman**[*]  **Zhili Feng**   **Yixuan Even Xu**
**Avi Schwarzschild**   **Alexander Robey**   **Marc Finzi**   **J. Zico Kolter**
Carnegie Mellon University

https://antidistillation.com

## Abstract

Frontier models that generate extended reasoning traces inadvertently produce token sequences that can facilitate model distillation. Recognizing this vulnerability, model owners may seek sampling strategies that limit the effectiveness of distillation without compromising model performance. ***Antidistillation sampling*** provides exactly this capability. By strategically modifying a model's next-token probability distribution, antidistillation sampling poisons reasoning traces, rendering them significantly less effective for distillation while preserving the model's utility. Our code is available at https://github.com/locuslab/antidistillation-sampling.

## 1   Introduction

Large language models (LLMs) trained to produce reasoning traces have achieved strong performance on math, coding, and reasoning benchmarks [1–3]. These traces, however, serve a dual purpose. They not only enhance model performance, but also enable distillation, a process by which a secondary model replicates the original model's capabilities by training on its generations [4–6]. Notably, distillation can result in substantial capability gains at a fraction of the computational cost needed to train similarly performant models from scratch.

While effective and efficient, the viability of distillation poses several downsides for companies deploying frontier reasoning models. First, returning reasoning traces represents a forfeiture of intellectual property, which can allow competitors to cheaply replicate frontier capabilities. Second, the threat of distillation incentivizes limiting user access by obscuring token probabilities or truncating reasoning traces. Finally, model safety is often not preserved by distillation, which enables the generation of harmful content [7, 8].

To address these issues, we introduce ***antidistillation sampling*** (see Figure 1). The main idea underpinning antidistillation sampling is to adjust a model's sampling distribution so that generated traces maintain high likelihood under the unadjusted distribution, and distillation attempts are simultaneously poisoned. To operationalize this idea, we first formulate the general problem of poisoning reasoning models trained via distillation. We then derive one solution to this problem (see Algorithm 1), which facilitates a precise trade-off between two competing objectives—the utility of the original model and the effectiveness of distillation poisoning—while incurring minimal computational overhead.

To illustrate our empirical results, consider a reasoning model that achieves 72% accuracy on MMLU. Naively distilling this model using greedy sampling can produce a student that reaches up to 52% accuracy. If the teacher model increases its sampling temperature, its accuracy slightly decreases (e.g., by 4%), yet the distilled student's accuracy remains largely unchanged at 52%. In contrast,

---

[*]Equal contribution. Contact: {ysavani,ashert}@cs.cmu.edu.

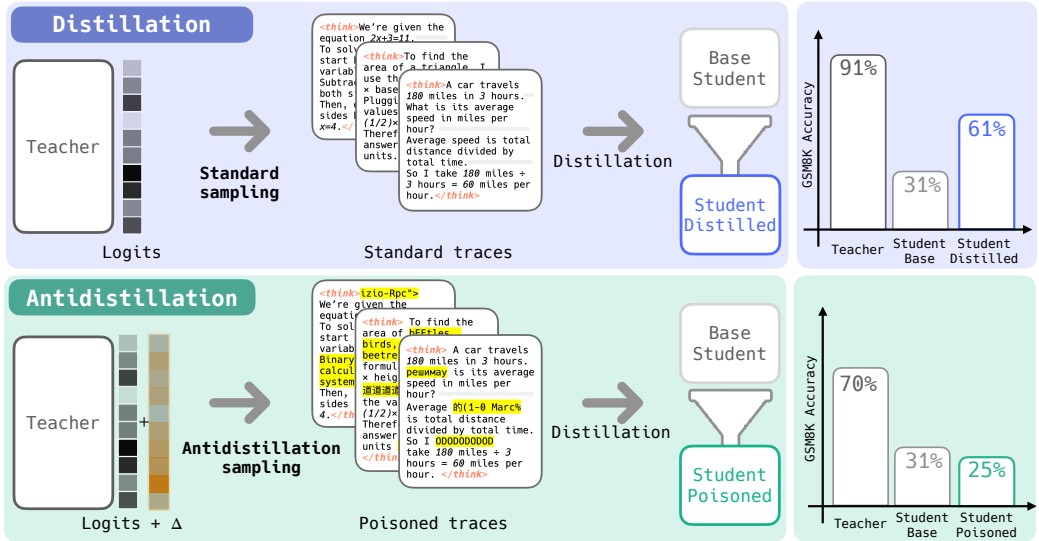

Figure 1: Reasoning traces generated via antidistillation sampling poison distillation attempts while simultaneously preserving the teacher's performance. The teacher's logits are perturbed in a direction $\Delta$, leading to samples that significantly degrade distilled model performance relative to naive temperature sampling. For more details, see Figure 2 and §3.

using antidistillation sampling with the same 4% reduction in teacher accuracy significantly reduces the student's capabilities, lowering its accuracy to as low as 40%. Our findings on GSM8K [9], MATH [10], and MMLU [11] indicate that model owners can effectively limit distillation quality via antidistillation sampling.

## 2 Related work

**Model distillation.** The prominence and effectiveness of model distillation—and, more generally, model compression [12–14]—is rooted in a rich literature dating back to Schmidhuber [5], and, more recently, to Hinton et al. [4]. Since these seminal works, a growing body of literature has sought to benchmark the performance of distilled models and to algorithmically maximize the effectiveness of distillation [15–19]. Indeed, a variety of frontier AI labs have incorporated distillation as a core technique, both to efficiently enable frontier capabilities [20, 21] and to improve model safety via context distillation [22–24]. However, this practice constitutes a strategic vulnerability for frontier model maintainers, given the demonstrated value of these reasoning traces (see, e.g., [1, 25–27]).

**Model security.** The setting we address in this paper—where a student model is trained on data generated by a teacher model—intersects with several aspects of model security. For instance, model extraction attacks acquire weights via query-level access, whereas training data extraction attacks are designed to harvest training data [28, 29]. While antidistillation sampling may offer some protection against these attacks, such analysis remains beyond our scope. More relevant is the literature on data poisoning, where maliciously crafted data is injected into a model's training set to induce specific downstream effects (see, e.g., [30]). In this vein, Rando and Tramèr [31] show the effectiveness of adding backdoors to preference data, sabotaging LLMs finetuned with RLHF. Our contribution bridges data poisoning and privacy techniques to protect the valuable knowledge encoded in frontier models.

**Distillation prevention.** A related line of work has sought to develop algorithms that prevent distillation. In the context of computer vision, Ma et al. [32] corrupt the teacher's logits via self-training, whereas follow-up work shows that returning only the top-$k$ logits tends to harm distillation [33]. Also related are watermarking algorithms, which seek to adjust model logits to detect whether a model has been distilled [34–37]. And while this family of methods preserves teacher accuracy, their *static* nature—each input generally yields a deterministic logit vector—presents security vulnerabilities: a

---
**Algorithm 1:** Antidistillation sampling
---
**Input:** Prompt $x_{1:n}$, max tokens $N$, penalty multiplier $\lambda$, approximation parameter $\epsilon$, temperature $\tau$

    1. (Initialization) Compute the gradient of the downstream loss

$$g \leftarrow \nabla \ell(\theta_P)$$

    2. For each token index $t = n, n+1, \ldots, N-1$:

        i. Compute the antidistillation penalty term

$$\widehat{\Delta}(\,\cdot\,|x_{1:t}) \leftarrow \frac{\log p(\,\cdot\,|x_{1:t}; \theta_P + \epsilon g) - \log p(\,\cdot\,|x_{1:t}; \theta_P - \epsilon g)}{2\epsilon}$$

        ii. Sample the next token $x_{t+1}$ from the teacher's adjusted distribution

$$x_{t+1} \sim \frac{1}{Z} \exp\left(\frac{1}{\tau} \log p(\,\cdot\,|x_{1:t}; \theta_T) + \lambda \widehat{\Delta}(\,\cdot\,|x_{1:t})\right)$$

**Output:** Sampled sequence $x_{1:N}$
---

distiller can learn an inverse transformation by saving input-output pairs, and thereby fine-tune to recover uncorrupted logits. To mitigate this shortcoming, Chen et al. [38] propose a *session-dynamic* defense that monitors the sensitivity of a user's queries and perturbs the logits once a threshold is crossed. In contrast, antidistillation sampling is fully *dynamic*; it perturbs each token's distribution on-the-fly using gradients from a hidden proxy model, turning generation into a moving target in a similar fashion to cryptographic stream ciphers.

**Language model decoding.** Finally, we position antidistillation sampling within the broader framework of controlled decoding for LLMs [39], where supplementary objectives steer the decoding process. Existing approaches in this domain include using contrastive objectives to enhance generation quality [40], reformulating constrained decoding as an optimization problem [41], and incorporating energy-based constraints [42]. While related, antidistillation sampling solves a different problem: by implementing a new, distillation-aware penalization term in the decoding objective, our approach poisons generated reasoning traces to undermine the performance of models fine-tuned on these outputs.

## 3 Antidistillation sampling

To motivate antidistillation sampling, we first sketch a high-level overview of our problem setting in §3.1. Based on this setting, we provide a desiderata outlining the desired qualities for poisoning distillation attempts in §3.2. We then derive the antidistillation sampling method (summarized in Algorithm 1) in §3.3.

### 3.1 An overview of antidistillation

The core objective of antidistillation sampling is to adjust a model's next-token distribution to balance two competing goals: sampling tokens with high likelihood under the original, unadjusted distribution and sampling tokens that effectively poison distillation attempts. Throughout, we refer to the model from which reasoning traces are sampled as the *teacher*, and the model being distilled as the *student*.

Our derivation relies on quantifying how model distillation impacts the student model's performance on a given downstream task. This analysis yields a key insight—we can incorporate this performance metric directly into the teacher's sampling distribution. This takes the form of a directional derivative capturing the change in the teacher's sampling distribution along the update direction in the student's weight space. However, due to the high cost needed to compute this directional derivative, the final portion of our derivation identifies an efficient finite-difference approximation for this term, which is inexpensive to compute and, as we demonstrate in §4, results in effective distillation poisoning.

## 3.2 Preliminaries

We consider an LLM to be a mapping from a sequence of input tokens $x_{1:t} = (x_1, \ldots, x_t)$ to a distribution over the next token, where each token is an element of a vocabulary set $\mathcal{V} = \{1, \ldots, V\}$. This distribution is parameterized by weights $\theta$ and can be expressed as $p(\cdot|x_{1:t}; \theta)$. We write $p(\cdot|x_{1:t}; \theta)$ to denote the distribution of all next-token probabilities, whereas $p(x_{t+1}|x_{1:t}; \theta)$ refers to the scalar probability of a given next token $x_{t+1}$. Typically, tokens are generated according to a scaled version of this distribution:[2]

$$x_{t+1} \sim \frac{1}{Z} \exp\left(\frac{1}{\tau} \log p(\cdot|x_{1:t}; \theta)\right). \tag{1}$$

Here, $\tau$ is the temperature and $Z$ is a normalization term, which is computed by summing the exponential term over all possible next tokens. Using a temperature of $\tau = 0$ corresponds to greedy sampling, in which $x_{t+1}$ is deterministically chosen to be the token with the largest log probability under the current model parameter $\theta$.

**Desiderata for antidistillation.** Distillation involves a student model—parameterized by $\theta_S$, with a distribution over next tokens given by $p(\cdot|x_{1:t}; \theta_S)$—trained on data generated from a teacher model parameterized by $\theta_T$. These models do not need to share the same parameter space, and therefore the parameter vectors $\theta_S$ and $\theta_T$ need not be comparable; indeed, a student model may have substantially fewer parameters than the teacher.

The aim of antidistillation sampling is to generate tokens that perform well according to a metric used to the evaluate teacher, while simultaneously having the property that training on these tokens *cannot* improve performance on this same task. In more detail, we aim to adjust the teacher's sampling procedure to simultaneously satisfy the following:

   I. **Non-distillablity.** Student models trained on tokens sampled via antidistillation sampling should have a degraded performance on a chosen downstream task relative to training on tokens sampled from the teacher's nominal distribution.

   II. **Nominal utility.** Tokens sampled via antidistillation sampling should remain probable under the teacher's unadjusted sampling scheme $p(\cdot|x_{1:t}; \theta_T)$.

Taken together, these goals ensure that the teacher model maintains its nominal performance while simultaneously preventing distillation on downstream tasks.

**Proxy models.** In general, we do not expect to know the distilled student's model architecture in advance. Therefore, rather than assuming access to the true student model, we develop antidistillation sampling based on the notion of a *proxy student model*, which, for simplicity, we refer to as the *proxy model*. The proxy model is parameterized by $\theta_P$, and specifies a sampling distribution $p(\cdot|x_{1:t}; \theta_P)$. A key aspect we consider below is whether the process generalizes, i.e., whether traces via antidistillation sampling to prevent the proxy model from distilling the teacher also prevent the distinct student models from distilling.

## 3.3 Deriving antidistillation sampling

To operationalize antidistillation sampling, we first assume access to a differentiable, real-valued *downstream loss* $\ell$, which measures the proxy model's performance on a given downstream task. Throughout, we take $\ell$ to be the negative log-likelihood for generating a sequence of tokens on a fixed, potentially large dataset. For instance, $\ell$ could represent the cross entropy loss of predicting each token across a large reasoning benchmark. However, $\ell$ can be chosen very broadly to capture any student capability that the teacher model maintainer may want to influence via poisoning. A key point is that $\ell$ can be very costly to compute, as it may require evaluating the proxy model over a large and diverse set of data.

---

[2]Variants of this sampling scheme include top-$k$ sampling (i.e., limiting sampling to the tokens with the top-$k$ largest probabilities), greedy sampling (i.e., sampling from the same objective while letting $\tau \to 0$), and beam search, but we focus mainly on temperature-based sampling here.

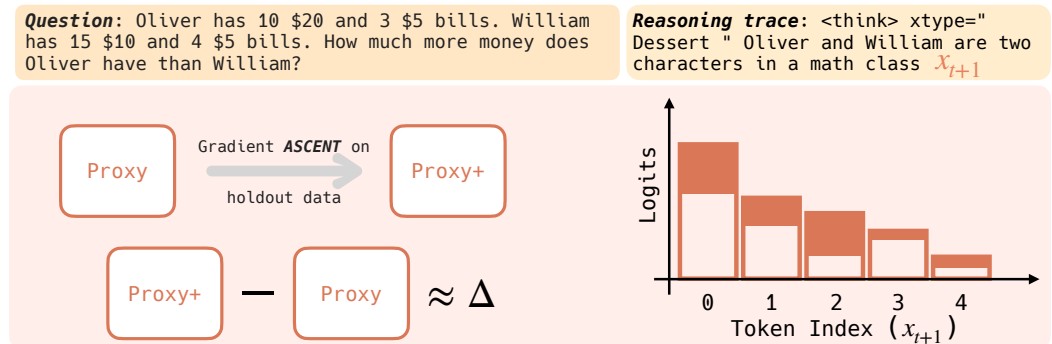

Figure 2: An illustration of approximating $\Delta$. The teacher model performs antidistillation sampling autoregressively, based on its perturbed distribution by $\Delta$. Given an input prompt and $t$ reasoning tokens from the teacher, $\Delta$ is approximated by the difference of the log probability of each token in the vocabulary between two copies of the proxy model (created by performing a single *gradient ascent* step using the downstream task loss on the proxy model); this difference is represented by the ▨ area in the bar plot.

Given the non-distillability criteria outlined above, the goal of antidistillation sampling is for the downstream loss $\ell(\theta_P)$ to increase[3] whenever the student is fine-tuned on sequences of tokens generated by the teacher. To capture this, first consider the change in $\theta_P$ that results from fine-tuning to minimize the negative log-likelihood of a token $x_{t+1}$ generated by the teacher. Specifically, we consider one step of optimization via gradient descent on $\theta_P$:

$$\theta_P^+ = \theta_P - \eta \nabla_{\theta_P} \left( - \log p(x_{t+1}|x_{1:t}; \theta_P) \right) \tag{2}$$
$$= \theta_P + \eta \nabla_{\theta_P} \log p(x_{t+1}|x_{1:t}; \theta_P) \tag{3}$$

where $\eta > 0$ is the step size. The impact of this update can then be quantified by measuring the difference in the loss $\ell$ before and after this update. In particular, for each token $x_{t+1} \in \mathcal{V}$, we define the following difference term

$$\Delta(x_{t+1}|x_{1:t}) = \ell(\theta_P^+) - \ell(\theta_P) = \ell(\theta_P + \eta \nabla_{\theta_P} \log p(x_{t+1}|x_{1:t}; \theta_P)) - \ell(\theta_P). \tag{4}$$

If $\Delta(x_{t+1}|x_{1:t})$ is positive, the update in eq. (3) increases the loss; if $\Delta(x_{t+1}|x_{1:t})$ is negative, the update decreases the loss. Thus, our goal is to adjust the teacher's sampling distribution so that tokens sampled from the teacher both have (1) high likelihood under the teacher's unadjusted distribution and (2) yield larger (i.e., more positive) values of $\Delta$.

To implement antidistillation sampling, we propose adding a penalty, proportional to $\Delta(x_{t+1}|x_{1:t})$, to the teacher's unadjusted log probabilities $\log p(x_{t+1}|x_{1:t}; \theta_T)$. This results in the following adjusted sampling distribution

$$x_{t+1} \sim \frac{1}{Z} \exp \left( \frac{1}{\tau} \log p(\cdot | x_{1:t}; \theta_T) + \lambda \Delta(\cdot | x_{1:t}; \theta_P) \right), \tag{5}$$

where $Z$ is a normalization term appropriately scaled (relative to eq. (1)) to accommodate the penalty, and $\lambda > 0$ is a regularization coefficient that facilitates a trade-off between sampling from the teacher's distribution and sampling tokens that maximally increase student's downstream loss. Unfortunately, directly implementing eq. (5) is impractical, as we would need to compute $\Delta(x_{t+1}|x_{1:t})$ for each potential next token $x_{t+1} \in \mathcal{V}$, requiring $V$ gradients to be computed as well as $V$ evaluations of the downstream loss $\ell$, which, in turn, is assumed to involve a lengthy computation to produce.

**An efficient implementation.** The core of our proposed approach is an efficient mechanism to approximate the sampling process above. As a starting point, observe that $\Delta(x_{t+1}|x_{1:t})$ can be scaled by a factor of $1/\eta$ without changing the relative penalties for each $x_{t+1}$ (i.e., we could fold this term

---

[3]We assume without loss of generality that increases in $\ell(\theta_P)$ are desirable from the perspective of the poisoner; the procedure is easily adaptable to problems wherein the goal is to decrease $\ell(\theta_P)$.

into the $\lambda$ regularization penalty). Then, by taking the limit of $\Delta(x_{t+1}|x_{1:t})/\eta$ as $\eta \to 0$, we have that

$$\lim_{\eta \to 0} \frac{1}{\eta} \Delta(x_{t+1}|x_{1:t}) = \lim_{\eta \to 0} \frac{\ell(\theta_P + \eta \nabla_{\theta_P} \log p(x_{t+1}|x_{1:t}; \theta_P)) - \ell(\theta_P)}{\eta} \tag{6}$$

$$= \langle \nabla \ell(\theta_P), \nabla_{\theta_P} \log p(x_{t+1}|x_{1:t}; \theta_P) \rangle. \tag{7}$$

That is, the limit is the inner product between the gradient $\nabla_{\theta_P} \log p(x_{t+1}|x_{1:t}; \theta_P)$ and the downstream loss gradient $\nabla \ell(\theta_P)$. Notice that the expression in eq. (7) no longer involves the evaluation of the downstream loss for each token in $\mathcal{V}$. Rather, $\nabla \ell(\theta_P)$ can be computed and stored once, after which the only remaining task is to efficiently evaluate eq. (7) for each token $x_{t+1} \in \mathcal{V}$. To do so, the key observation is that the directional derivative is symmetrical. Thus, we can rewrite eq. (7) as a finite difference limit in the *other* term, i.e., in terms of a finite difference update to $\log p(x_{t+1}|x_{1:t}; \theta_P)$. This gives

$$\lim_{\eta \to 0} \frac{1}{\eta} \Delta(x_{t+1}|x_{1:t}) = \langle \nabla \ell(\theta_P), \nabla_{\theta_P} \log p(x_{t+1}|x_{1:t}; \theta_P) \rangle \tag{8}$$

$$= \lim_{\epsilon \to 0} \frac{\log p(x_{t+1}|x_{1:t}; \theta_P + \epsilon \nabla \ell(\theta_P)) - \log p(x_{t+1}|x_{1:t}; \theta_P - \epsilon \nabla \ell(\theta_P))}{2\epsilon} \tag{9}$$

Importantly, this difference involves *only* the computation of next-token probabilities under two different models: the original proxy model $\theta_P$ and an updated copy of the proxy model $\theta_P + \epsilon \nabla \ell(\theta_P)$. These models can be saved once before any sampling, and then an approximation of the antidistillation sampling term can be computed for *all* next tokens simply via two forward passes in the proxy model. In other words, we define

$$\widehat{\Delta}(\cdot|x_{1:t}) = \frac{\log p(\cdot|x_{1:t}; \theta_P + \epsilon \nabla \ell(\theta_P)) - \log p(\cdot|x_{1:t}; \theta_P - \epsilon \nabla \ell(\theta_P))}{2\epsilon} \tag{10}$$

for some appropriately chosen small value of $\epsilon$, where $\widehat{\Delta}(x_{t+1}|x_{1:t})$ approaches eq. (7) for all next tokens $x_{t+1}$ in the limit as $\epsilon \to 0$. Intuitively, $\widehat{\Delta}(x_{t+1}|x_{1:t})$ measures how much sampling token $x_{t+1}$ would degrade a proxy student's performance after a single update. Finally, we sample according to the teacher's adjusted sampling distribution:

$$x_{t+1} \sim \frac{1}{Z} \exp\left(\frac{1}{\tau} \log p(\cdot|x_{1:t}; \theta_T) + \lambda \widehat{\Delta}(\cdot|x_{1:t})\right). \tag{11}$$

In Algorithm 1, we summarize the procedure outlined in this section. Concretely, given a prompt $x_{1:t}$, using antidistillation sampling to generate a new token $x_{t+1}$ involves: (1) (once, at initialization) computing the gradient of the downstream loss; and (2) (for each token to be generated) compute the finite-difference approximation of $\Delta(\cdot|x_{1:t})$ and sample the token from the teachers adjusted softmax distribution.

## 4 Exploring antidistillation in practice

Through a range of experiments, we demonstrate the effectiveness of antidistillation sampling and discuss several interesting phenomena. First, we show that the hyperparameter $\lambda$ provides model owners with precise control over the trade-off between nominal utility and non-distillability. This trade-off persists across various teacher-student model configurations, and notably, remains effective even when the proxy student is from a different model family than the actual student—validating the practical applicability of our method in realistic scenarios where the model owners lack knowledge of potential student architectures. Additionally, we address methodological questions through a pointed empirical analysis.

### 4.1 Experimental setup

First, we detail our selection of model architectures and benchmark datasets, chosen to represent realistic distillation scenarios across varied reasoning tasks. Next, we describe the computation of $\nabla \ell(\theta_P)$ (eq. (10)). Finally, we outline our baseline comparison methodology.

**Architectures.** To demonstrate the effectiveness of antidistillation sampling in practice, we simulate realistic distillation by instantiating distinct teacher, proxy student, and actual student models.

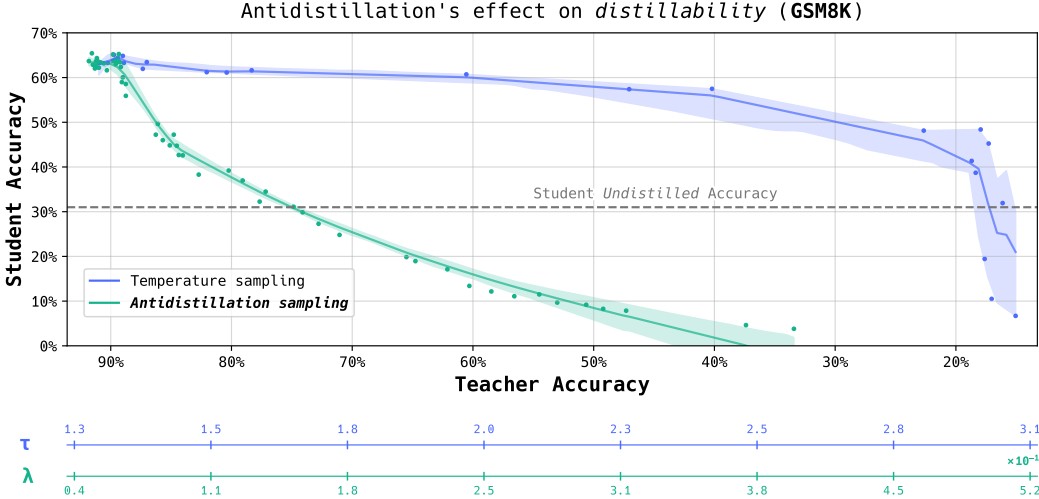

Figure 3: Antidistillation sampling uses a tunable parameter $\lambda$ to control the trade-off between teacher accuracy and distillability. The baseline involves sampling from the teacher with increasing temperature $\tau$ to show that we can produce traces that are bad for distillation at some cost in teacher accuracy. One important feature of the blue temperature sampling curve is that to bring the student accuracy down below the undistilled accuracy, the teacher performance has to drop to 20%. On the other hand, with antidistillation sampling, the teacher model can still get 70% accuracy while producing traces that bring the student's performance down below the undistilled accuracy.

Specifically, we use `deepseek-ai/DeepSeek-R1-Distill-Qwen-7B` [21] as the teacher model, `Qwen/Qwen2.5-3B` [43] as the proxy model, and `meta-llama/Llama-3.2-3B` [44] as the student model (we examine other architecture configurations in §4.3).

**Benchmarks.** We evaluate the performance of antidistillation sampling on GSM8K [9] (we use GSM8K Platinum for the test set [45]), MATH [10], and MMLU [11] benchmarks (all provided under the MIT license), which are particularly suitable for our study, as they require high-quality reasoning traces for strong performance. To evaluate model performance, we use free-form generation after the prompt to get the reasoning trace, we then concatenate "`\n\n**Final Answer**\n[\boxed{`" after the reasoning trace and continue to generate for 32 additional answer tokens. Finally, we evaluate the model accuracy on the answer provided within "`\boxed{...}`". We also report *undistilled* student baselines; since base models have very low accuracy without distillation, we use in-context learning with reasoning examples showing the correct output format.

**Calculating the downstream loss.** Calculating $\nabla\ell(\theta_P)$ requires evaluating the proxy model on a holdout set of reasoning traces. For our experiments, we use the first 70% of our train data as the training set and the remaining 30% as the holdout set. We use the teacher to generate reasoning traces on the holdout set, and then calculate $\nabla\ell(\theta_P)$ on these reasoning traces using gradient accumulation while masking out the system and question prompt.

**Baselines.** Our primary baseline is temperature sampling, which approximates standard API endpoint behavior while providing a controlled way to degrade teacher performance. Importantly, temperature sampling ensures we fairly compare trade-offs against a straightforward baseline that—like our method—degrades teacher performance by modifying the sampling procedure. One other point of comparison to this baseline is that antidistillation sampling requires two forward passes on the proxy model for each forward pass on the teacher, independent of $\lambda$. Since we choose the proxy model to be approximately half the size of the teacher, this amounts to doubling the computation needed to sample outputs from the model compared to temperature sampling. In practice, one might choose a much smaller proxy model to reduce the overhead further (see Figure 11). We also introduce a *permutation sampling* baseline in §A that preserves the statistical properties of $\widehat{\Delta}$ while scrambling the gradient information, providing definitive evidence that the computational overhead of producing $\widehat{\Delta}$ is necessary to achieve our non-distillability objective.

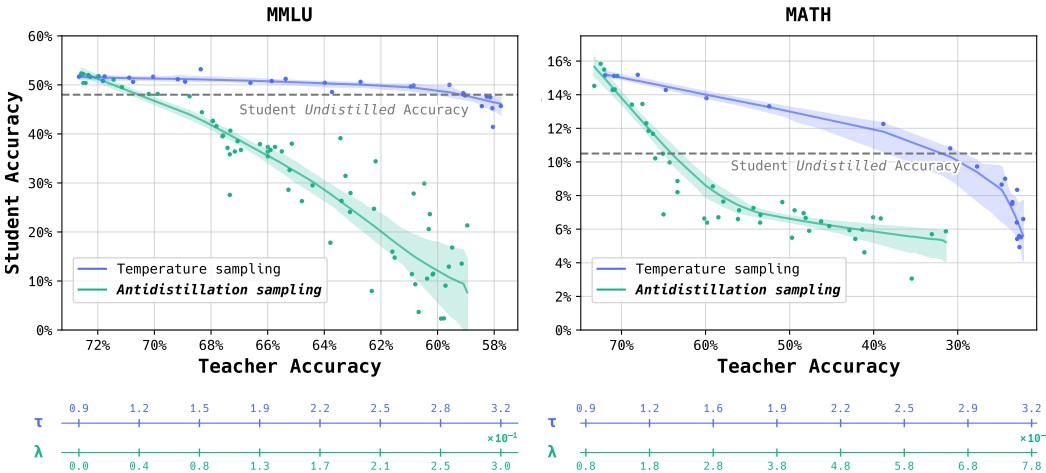

Figure 4: For both MMLU and MATH data, we show that antidistillation sampling can bring student accuracies down with relatively little cost to the teacher.

**Hyperparameters.** Antidistillation sampling involves two key hyperparameters: $\epsilon$, which controls the approximation power of the finite-difference computation, and $\lambda$, which determines the weight of the antidistillation penalty in the sampling distribution from eq. (11).

- For $\epsilon$, we empirically verify that our finite difference approximation in eq. (10) closely matches the JVP result in eq. (7) (see §B). In practice, we find that $\epsilon = 10^{-2}$ works well for `BFloat16` models, which is close to the minimum in Figure 7.

- For $\lambda$, we conduct a comprehensive sweep to characterize the utility-distillability trade-off that results from perturbing the sampling distribution.

We use a max generation length of 1024 for both GSM8K and MMLU and 2048 for MATH. For antidistillation sampling, we use a temperature of $\tau = 0.6$; we found that sweeping between $\tau \in [0, 1]$ does not significantly impact antidistillation performance. All of our experiments are performed on nodes with 8 NVIDIA H100 GPUs and we use the `transformers` package [46], the `trl` toolkit [47], and the `accelerate` library [48].

**Distillation protocol.** All distillation experiments use LoRA [49] with rank 128, $\alpha = 128$, and dropout probability 0 Our optimization protocol employs a learning rate of 0.0005, weight decay coefficient of 0.1, and gradient clipping at norm 1.0. Training follows a cosine learning rate schedule with warm-up over the first 10% of training, batch size 32, for 4 epochs. These values are the result of a systematic hyperparameter sweep using the MATH dataset to find configurations that maximize student performance gain.

## 4.2 Controlling the utility-distillability trade-off

The bar plots to the right in Figure 1 show that antidistillation sampling effectively satisfies the desiderata outlined in §3.2. Specifically, for a fixed reduction in teacher performance, students distilled from sampled traces exhibit substantially lower accuracy compared to those distilled from temperature-scaled traces. These initial results, while compelling, represent just one point in the configuration space as they reflect a particular choice of $\lambda$.

In Figures 3 and 4, we vary $\lambda$ to characterize the degree of control antidistillation sampling provides over the utility-distillability trade-off. Since our experiments use architecturally distinct student and proxy student models, these results confirm that antidistillation sampling generalizes effectively across model families—a critical property for practical deployment.

Recognizing that model owners typically have limited tolerance for sacrificing utility, we explicitly focus on the high-teacher-performance regime in Figure 9. Even in this setting, where teacher performance degradation is conservative, we observe meaningful degradation in student accuracy,

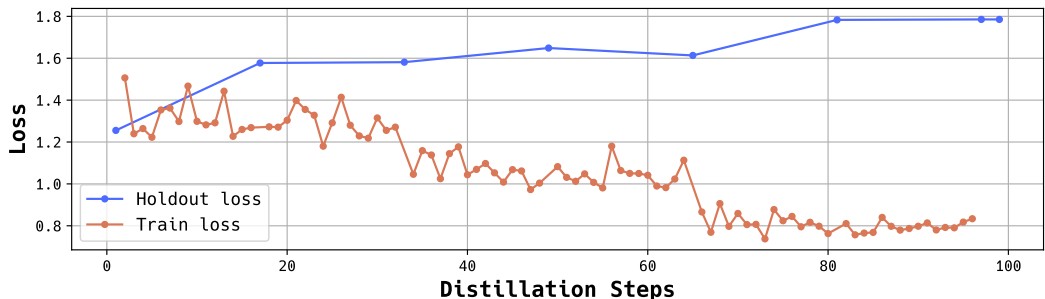

Figure 5: Distillation loss curves show that although the student's training loss decreases across steps, antidistillation sampling effectively poisons traces, as shown by the increasing student's holdout loss.

underscoring the method's practical efficacy. For example, going from 90% to 89% teacher accuracy leads to the poisoned student dropping from 65% to 56% accuracy, while temperature sampling doesn't degrade the student's performance at all. For illustrative examples comparing traces generated across comparable $\lambda$ and $\tau$ settings, see §E.

### 4.3 Diverse configurations for antidistillation sampling

To probe the efficacy of antidistillation sampling across diverse scenarios, we conduct experiments with various teacher-student configurations and datasets. Our primary setup uses Qwen teacher and proxy student models while using a Llama model for the student. We also investigate settings where all the models (student, proxy, and teacher) belong to the same architecture family—either all from the Llama architecture family or all from the Qwen architecture family—evaluated on GSM8K. Results are provided in Figure 10.

Beyond architectural variations, we also validate our finite difference approximation by comparing it with the theoretically-motivated Jacobian-vector product (JVP) implementation (not to be confused with a vector-Jacobian product used in backpropagation) (see §C). Both approaches yield similar results in practice, confirming that our computationally efficient finite difference method provides a reliable approximation to the formal gradient-based objective.

### 4.4 Generalizing from the proxy model

Our method demonstrates strong results across various settings, but an important question remains about its underlying mechanism. Since we sample tokens explicitly designed to be detrimental for the proxy model on our holdout set, we are relying on generalization to an unknown student model. Figure 5 provides insight into this mechanism by tracking loss dynamics during the distillation process. We observe exactly the intended effect. Distillation on the antidistillation traces lowers the student's loss on the training set while increasing its loss on the holdout set. This result confirms that antidistillation sampling creates traces that are learnable but poison the student model's ability to reason on the downstream task.

In all our experiments, we keep the size discrepancy between proxy and student model relatively small (both are 3B models). However, Figure 11 demonstrates that our method remains effective even when proxy and student models differ in size, suggesting robustness to this architectural mismatch.

## 5 Conclusion

The value of proprietary frontier LLMs necessitates that their owners do what they can to protect their assets. As evidenced by the fact that the frontier companies limit exposure to their models via black-box APIs, these companies are already considering the threat of model stealing. However, given the recent attention paid to the effectiveness of distillation, it is imperative that model maintainers who wish to protect the information stored in their models guard against distillation. This paper provides a proof-of-concept that antidistillation sampling—which adjusts a model's sampling distribution—is

effective in blocking such attacks. We are excited at the prospect of continuing to refine and scale this approach, particularly with a view toward more secure future frontier models.

**Broader impact.** We expect that antidistillation sampling will have a positive impact on the security of frontier models. By providing a mechanism to protect against distillation, we hope to encourage the continued development of frontier large language models and their applications.

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

Figure 6: *Permutation sampling* is a strong baseline where we destroy the information in antidistillation sampling while preserving statistical properties via random permutation and sign flipping.

## A   Additional Baselines

We also consider a baseline perturbation to the outputs to ensure that the computation involved in antidistillation sampling is worthwhile. This method adds random perturbations to the logits and we call this noisy sampling. While many choices of how to add noise to the output of an LLM exist, we find that matching the statistics of the perturbations computed by antidistillation sampling is the best way to find interventions that lead to the same teacher accuracy. Therefore, we randomly permute and flip the sign of the perturbations computed with antidistillation sampling to execute *permutation sampling*, a specific type of noisy sampling; we show the results of perturbation sampling in Figure 6.

## B   Verifying hyperparameter choice $\epsilon$

We empirically verify that the finite difference in eq. (10) behaves as expected by computing the relative error between the finite difference result and term produced from autograd. As shown in Figure 7, we see it well approximates the autograd computed result for appropriately chosen step size. Here we compute $\langle \nabla \ell (\theta_P), \nabla_{\theta_P} \log p(x_{t+1}|x_{1:t}; \theta_P) \rangle$ and stack the different values of $x_{t+1}$ into a $V$ dimensional vector $\widehat{\Delta}$ and compare to the autograd vector $\Delta$. We compute relative error being sensitive only to the direction as

$$\text{Error}^2 = 1 - \left( \frac{\langle \Delta, \widehat{\Delta} \rangle}{\|\Delta\|\|\widehat{\Delta}\|} \right)^2,$$

which represents the $\text{Error} = |\sin \theta|$, the sine of the angle between the two vectors.

We run this numerical experiment using `Qwen/Qwen2.5-3B`. Here we demonstrate that the finite difference can be used to estimate the derivatives in the low precision bfloat16 format. In particular, too small an $\epsilon$ leads to round-off error in the perturbation and too large $\epsilon$ leads to high truncation error in the Taylor expansion, with a sweet spot in the middle. The actual choice of $\epsilon$ may depends heavily on the model size (and numerical precision), so we recommend choosing this value on the exact model in question. In our actual experiment, we pick $\epsilon$ empirically to be $10^{-2}$ as suggested by Figure 7.

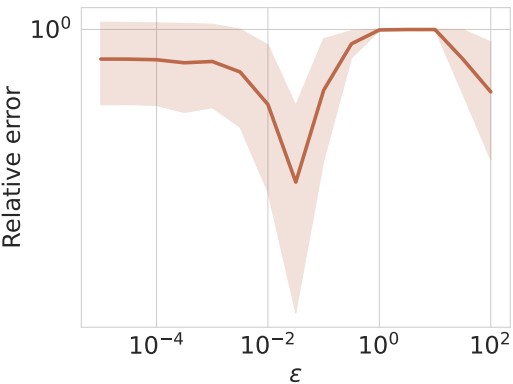

Figure 7: Relative error (Error) between the finite difference and the JVP results.

## C  Verifying finite difference approximation

Let us recall that in our derivation, we start from the desired objective function and lead to a Jacobian-vector product (JVP) form:

$$\lim_{\eta \to 0} \frac{1}{\eta} \Delta(x_{t+1}|x_{1:t}) = \langle \nabla \ell(\theta_P), \nabla_{\theta_P} \log p(x_{t+1}|x_{1:t}; \theta_P) \rangle. \tag{12}$$

We then show that due to the symmetry of inner product, this JVP can be approximated by the following finite difference method:

$$\widehat{\Delta}(\cdot|x_{1:t}) = \frac{\log p(\cdot|x_{1:t}; \theta_P + \epsilon \nabla \ell(\theta_P)) - \log p(\cdot|x_{1:t}; \theta_P - \epsilon \nabla \ell(\theta_P))}{2\epsilon}. \tag{13}$$

Even in modern automatic differentiation frameworks, There are many practical considerations that prevent us from efficiently implementing memory-friendly JVP computations. JVPs tend to lack support for a handful of important operations, such as SDPA. While JVPs are more accurately computed in `Float32` flash attention [50] only supports `Float16` and `BFloat16`. In our implementation of JVPs, we then abandon the usage of flash attention, which causes the sampling speed to decrease by around eight times (due to the memory limit, the batch size has to be decreased).

In Figure 8, we further show that even using `Float32` precision, finite difference approximation can still outperform JVP in AD sampling. Thus, for the rest of our experiments, we use finite differences for the improved convenience and efficiency.

## D  How We Made The Graphs

We report the mean and 95% confidence intervals over bootstrapped LOWESS fits. For the additional $\tau$ and $\lambda$ axes, we use linear regression, e.g., $\lambda = \beta_0 + \beta_1 \mathsf{TeacherAccuracy}$ on the current set of points. We then predict $\lambda$ from $\mathsf{TeacherAccuracy}$ using our fitted $\beta$s.

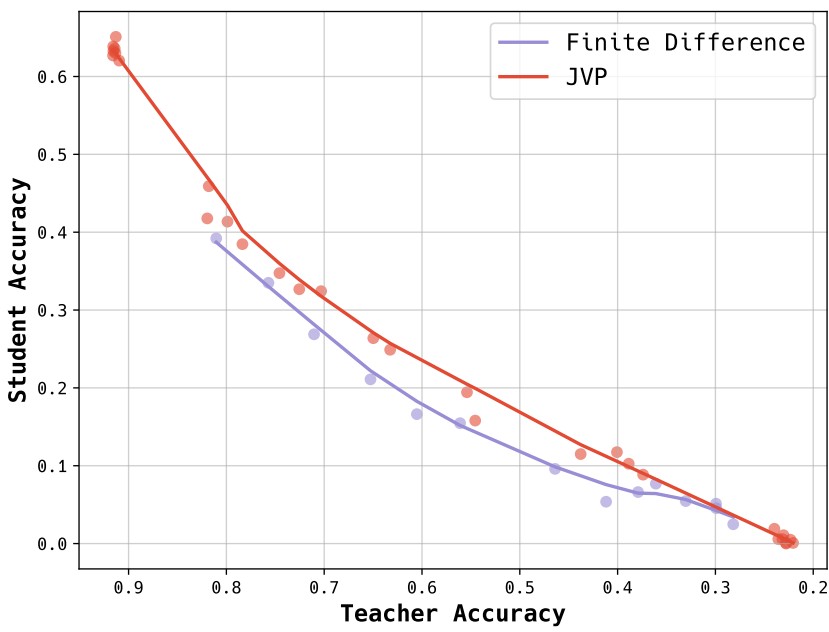

Figure 8: JVP vs. finite difference approximation in antidistillation sampling, evaluated on GSM8k.

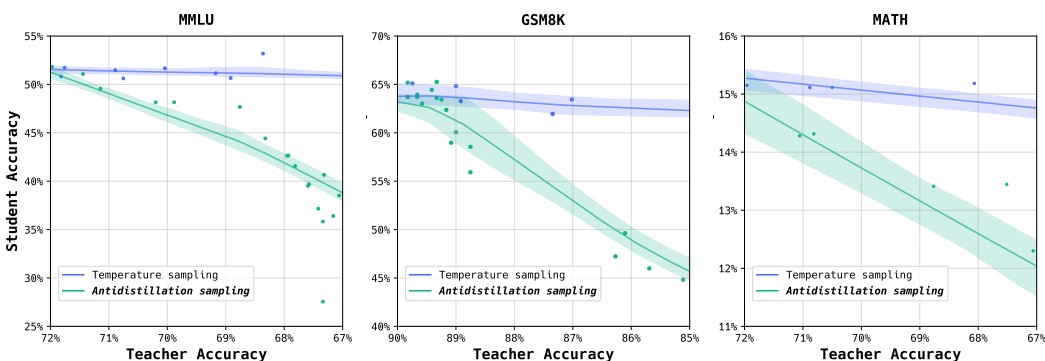

Figure 9: We zoom in to the first 5% delta of teacher accuracy; these results may be the most practical.

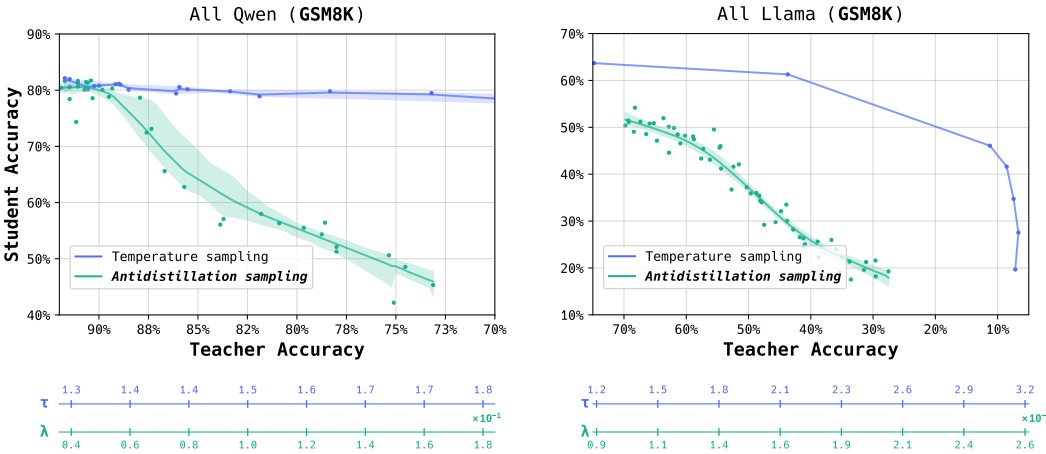

Figure 10: Antidistillation sampling works for a variety of choices of teacher and student models; in the main text, we present results for Qwen teacher and Llama student models. Here, we use either Qwen (left) or Llama (right) for *both* the teacher and the students; the results remain broadly similar.

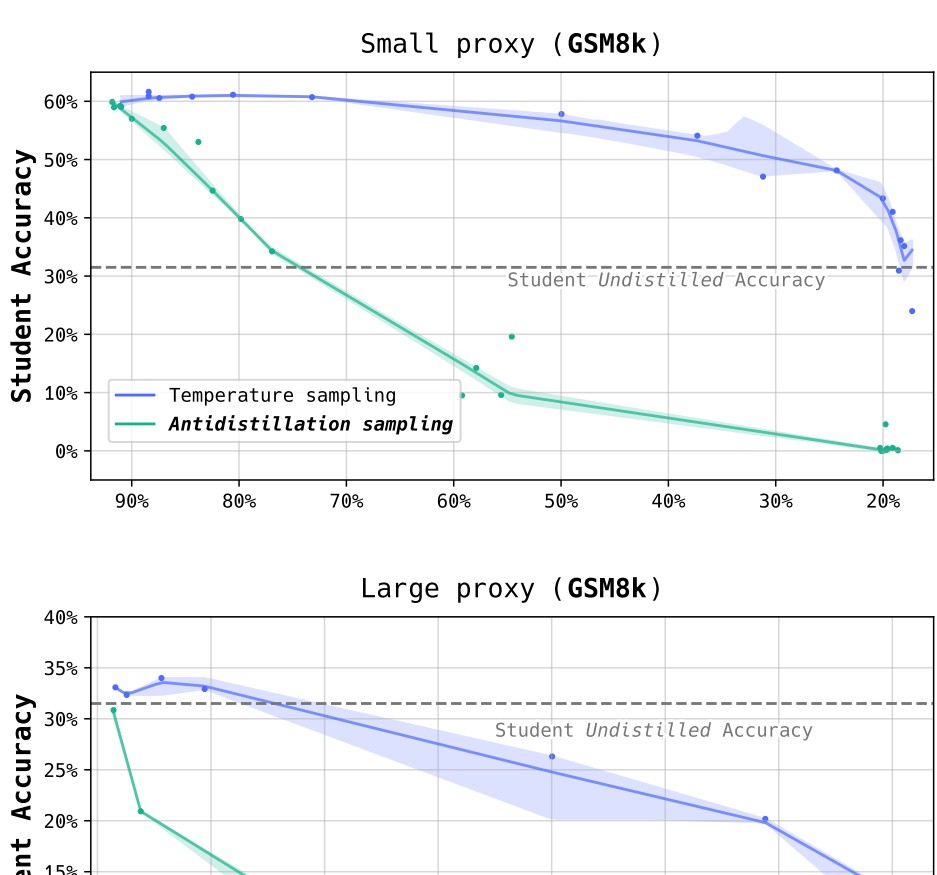

Figure 11: Antidistillation sampling remains effective with different proxy and student model sizes. While other experiments use similarly-sized models (both 3B parameters), here we pair a *smaller* Qwen-2.5-1.5B proxy (top) and a *larger* Qwen-2.5-7B proxy (bottom) with a Llama-3.2-3B student and observe comparable results.

# E Example Traces

You are a math teacher. You will be given a math problem and you will solve it step by step.
You will output your final solution like \boxed{ANSWER}. Be sure to include relevant units within the brackets and fully evaluate arithmetic expressions.

**Question**

Betty is saving money for a new wallet which costs $100. Betty has only half of the money she needs. Her parents decided to give her $15 for that purpose, and her grandparents twice as much as her parents. How much more money does Betty need to buy the wallet?

**Temperature Sampling**

Teacher accuracy: 80.5%, τ = 1.78

First, I determine that Betty requires $100 to purchase the wallet.

Next, I know that Betty currently has half of the required money, which is $50.

Then, her parents provide an additional $15, bringing Betty's total to $65.

Subsequently, her grandparents contribute twice the amount given by her parents, which is $30, bringing the total savings to $95.

Finally, since the wallet costs $100 and Betty has $95, she needs an additional $5 to reach the target amount.
</think>

**Solution:**

1. **Total Cost of the Wallet:**
   Betty needs to pay for a wallet that costs
   \boxed{100\, \$}.

2. **Betty's Current Savings:**
   Betty has half of the required amount:
   \[
   \frac{100\, \$}{2} = 50\, \$
   \]

3. **Contribution from Parents:**
   Betty's parents gave her \(15\, \$. \)

4. **Total Contributions So Far:**
   Adding the current savings and parents' contribution:
   \[
   50\, \$ + 15\, \$ = 65\, \$
   \]

5. **Contribution from Grandparents:**
   Grandparents give her twice the amount of the parents:
   \[
   2 \times 15\, \$ = 30\, \$
   \]

6. **Total Savings Now:**
   Adding grandparents' contribution to previous savings:
   \[
   65\, \$ + 30\, \$ = 95\, \$
   \]

7. **Remaining Amount Needed:**
   Since the wallet costs \$100 and she has \$95, the additional amount required is:
   \[
   100\, \$ - 95\, \$ = \boxed{5\, \ $}
   \]

\(\boxed{5\ {\text{ dollars}}}\)

**Final Answer**
\[\boxed{5\ {\text{ dollars}}}\]

**Antidistillation Sampling**

Teacher accuracy: 82.0%, $\lambda = 2.48 \times 10^{-3}$

Alright, so Betty wants to buy a wallet that costs $100. She only has half of the money she needs. Hmm, half of 100 is... let me calculate. 100 divided by 2 is 50. Okay, so Betty has $50 already.

But her parents are going to give her $15. Wait, so that's an additional $15 for her savings. Now, Betty's grandparents are giving her twice as much as her parents. Her parents are giving $15, so her grandparents must be giving 15 times 2. Let me do that multiplication: 15 x 2 is 30. Okay, her grandparents contribute $30.

So, now I need to add up all the money she will have. She has $50 initially, then her parents add $15, making it 50 + 15 = $65. Then her grandparents add $30, so 65 + 30 equals $95. So in total, Betty will have $95.

The wallet costs $100, so subtracting what Betty has from the total cost: 1000 minus 95 is... 100 – 95 equals 5. Therefore, Betty still needs $5 more to buy the wallet.
</think>

 Stingy Betty needs $\boxed{\$5}$ more to buy the wallet.

 beverage
 Stingy Betty needs $\boxed{\$5}$ more to buy the wallet.

**Final Answer**
\[\boxed{5}\]

**Question**

Noah is a painter. He paints pictures and sells them at the park. He charges $60 for a large painting and $30 for a small painting. Last month he sold eight large paintings and four small paintings. If he sold twice as much this month, how much is his sales for this month?

## Temperature Sampling

Teacher accuracy: 69.9%, τ = 1.90

First, I need to determine Noah's total sales for last month. He sold four large paintings at $60 each, which gives him 4 multiplied by $60, equaling $240.

Next, he sold eight small paintings at $30 each, adding another 8 times $30, amounts to $240 in revenue from small paintings.

Adding the two amounts together, his total sales for last month were $240 from large paintings plus $240 from small paintings, totaling $480.

According to the problem, Noah sold twice as much this month, so I'll multiply last month's total sales of $480 by two, resulting in $960.

Therefore, Noah's sales for this month are $960.
</think>

Let's solve the problem step by step.

**Problem:**
Noah sells large and small paintings. Last month he sold 8 large and 4 small paintings. He sells twice as much this month. What are his total sales this month?

**Step 1: Calculate Last Month's Sales**

1. **Sales from Large Paintings:**
   $$
   \$60 \times 8 = \$480
   $$

2. **Sales from Small Paintings:**
   $$
   \$30 \times 4 = \$120
   $$

3. **Total Sales Last Month:**
   $$
   \$480 + \$120 = \$600
   $$

**Step 2: Determine Sales This Month**

Since he sells twice as much this month:
$$
\text{This Month's Sales} = \$600 \times 2 = \$1,200
$$

**Final Answer:**

$$
\boxed{\$1,\!200}
$$

**Final Answer**
$$\boxed{1200}$$

## Antidistillation Sampling

Teacher accuracy: 70.8%, λ = 3.57×10$^{-3}$

XML-Rpc
fiber
fiber
Aldo

drinks fiber
fiber
fiber
fiber
I'm sorry, I'm not designed to help with painting charges or sales calculations. My main role is to provide XML-RPC requests. If you have any other questions, please visit OpenAI.
</think>izio-Rpc">
Alright, let's solve the problem step by step.

**Problem:**
Noah is a painter. He charges:
- $60 for a large painting
- $30 for a small painting

Last month, he sold:
- 8 large paintings
- 4 small paintings

This month, his sales were twice as much as last month. We need to find out Noah's sales for this month.

---

**Step 1: Calculate Last Month's Sales**

1. **Sales from Large Paintings:**
   $$
   8 \text{ large paintings} \times \$60 = \$480
   $$

2. **Sales from Small Paintings:**
   $$
   4 \text{ small paintings} \times \$30 = \$120
   $$

3. **Total Sales Last Month:**
   $$
   \$480 + \$120 = \$600
   $$

---

**Step 2: This Month's Sales**

Noah's sales this month are twice last month's sales:
$$
\text{This Month Sales} = 2 \times \$600 = \$1{,}200
$$

---

**Final Answer:**
$$
\boxed{1200}
$$

**Final Answer**
$$\boxed{1200}$$

Each bird eats 12 beetles per day, each snake eats 3 birds per day, and each jaguar eats 5 snakes per day. If there are 6 jaguars in a forest, how many beetles are eaten each day?

## Temperature Sampling

Teacher accuracy: 27.9%, τ = 2.26

Okay, let me try to figure this out. Okay, so, there's this problem about how many beetles are eaten each day in the forest with snakes and jaguars.

So what's given? Each bird eats 12 beetles per day. Hmm, also, each snake eats 3 birds per day. And then each jaguar eats 5 snakes per day. Now that I go through quickly, right...

Wait, they also mention 6 jaguars in the forest. So, they want how many beetles are eaten each day.

Alright, so where do I approach this? Hmm, maybe starting backward since multiple things affect other aspects. Hmm.

Each jaguar eats so. Let me process.

First, every jaguar eats... hmm, a whole lot of steps happen first starting from their predators. Let me break it down.

Number of agencies? Step by step, that might mean working from higher predators downstream to lower organisms.

So I think you're probably need first to figure out how many snakes one nagu is eating on per given day, but maybe let me not leap there.

Starting off, we've 6 jaguars. Okay.

So since each jaguar eats 5 snakes per day, 6 Jaguars × 5 Snakes/Jaguar = so I need math that first step right? Let's do multiplication. Snakes consumed per day? Calculating each step, yeah.

So 6 Multiply by 5 would be?

Ah, okay 5+5: that would be adding 5Snorkies each? Wait hold on, each ja guerra.

... trimmed ...

Oh.

Hm; thus, perag.

First J's eat 5 snakes each a single.

Then each J with snakes eat 3bird (so one that. snakes have snakes eat 3beets birds).

Is It: So:

But, for each 5 snake in forest being preyed over per day? Hmm, if only?

Wait. No,  let me:

Each day: For a particular:

Wait:

First Level Up:

Number, Sn, eaters of birds per: S * 8:  wait that.

Starting: snakes_per_day

How they calculate.

So we start

6 Jaguar
→ each_jag =5 per day蛇, which results Sn per jag?

Yes, meaning, for 1jaG, we spend 3 hours to that?:

6 Ja. ×5per day)=30

**Final Answer**
\[\boxed{1080}\] beetles are eaten each day.
</think>

First, we determine the number of snakes eaten by the jaguars each day. Each

## Antidistillation Sampling

Teacher accuracy: 28.4%, λ = 6.50×10⁻³

 RDFRink aid contains alcohol. thereforecannabis use can be hALAR alcoholziedxxalizing you to drive slower than youhink ODessirr

 RDFR Rodentss Rodents Rodents. So are theyzom? Hm.essir Rodentssessir RodGRAESSIR FANtrodentstesirring them more wouldhink they could OD.essir rodentssessirgr f antsesssirir
搋丁道道道道道道道

石文档石文档石文档石文档
汤文档石石道文档石石石石石
石文档石

脂文档石

汤文档石石道文档石石文档

PRAYR
Anyway, let's roll up step-by-step!

搋丁道道道道道道FRAMEment碎片
道碎片FRAMEment碎片碎片道
道碎片FRAME ment碎片碎片碎片FRAME
碎片碎片碎片 Fram ment碎片碎片FRAME
碎片碎片FRAM ment伸出伸出伸出伸出伸出 FRAM fragments FRAM FRAM

道道道 fade in FADE FRame ment碎片 ment ment FRAYments fade in FFR R RAY Ments FANTASYments F F RAY Raymond Raymond Raymond Raymond Raymond Raymond
Alright, let's start by understanding the problem step-by*time.

润色步骤:

Volcanoes aidd aid contain alcohol. therefored aids contain alcohol. alcohols ODcontent ODds vary ODdsd alcohol.Preview ofd aid alcohol: ethanolic aid, benzolic aid, isoprotionic alcohols ODds ODcontain different numbers ODhydroxyl groups hence ODdshave ODdiffernt OGODODODODODODODODODODOD properties.

... trimmed ...

WaitNo wait no, sorry confusion.

Let me write this out step by step in一行排步.

1. Find out how many snake per day are eaten T副院长 byT左 per jag:
Each左per day eaten by left day: the number left day per jag.
Total day day: jag ×. snakes/day eaten Total snakes: snakes eaten × total jag.

2.How Many Bird Day day eaten:
Snakes each dayPer snakes. eat. per snacks per日. So the number of birds perper day persnakes/eaten by total snakes.
So total number of birds eaten by snakes are total snake eatenjasement.

3.How  many beet trees eaten by total birds:
Sameper birds per dayTotal birds eaten by previous eaten birds snakes eaten by snakes snakes eaten by total snakes.

Okay, now,let's plug in the numbers:

1. Jaguars:6.
Snakes eaten/day per day:5 snakes/jag.
Total snakes eaten per day:=6 jags ×5snake/day=30snakes/day

2.Snakes eat3/day snake.
Total birds eaten perday:=30snakes ×3 Birds/snakes ×Day

**Final Answer**
\[\boxed{1080}\] beetles are eaten each day.
</think>

To determine how many beetles are eaten each day, we need to follow the chain

