# OpenReview forum: "Antidistillation Sampling"
_NeurIPS.cc/2025/Conference — NeurIPS 2025 poster_

### Official Review · Reviewer_PaZ6 · 2025-06-30

**Clarity:** 3
**Significance:** 3
**Originality:** 3
**Rating:** 4
**Confidence:** 3

**Summary:**

Extended reasoning traces are useful for model distillation. Therefore sampling strategies that prevent distillation are desirable. The authors present a sampling approach to enable this capability during inference time by modifying a model's next token distribution. This sampling poisons the reasoning trace making it difficult for distillation but still preserving accuracy of the teacher.

**Questions:**

1. GSM8K and Math are fairly saturated domains and distillation approaches have been shown to be effective for more challenging math domains such as AIME 2025, where long reasoning traces are required (32K or more). Could you share how your sampling approach would transfer to this setup, where there may be a tradeoff for student and teacher accuracy might be different as the number of tokens you need to solve a problem increases?
2. Can approaches such as Verifree[1] that increase log probs on only the solution still work well with the solution (and no thinking) from the poisoned traces?
3. How does the size of the proxy and student models interact? Do you see differences with efficacy of the poisoning over different model scales?
4. Would it be possible to do a human study comparing the utility of the answers with and without antidistillation sampling as the model seems to inject text within the solution that may reduce the utility of the solutions as seen in Appendix D

References
1. Reinforcing General Reasoning without Verifiers

**Ethical Concerns:**

["NO or VERY MINOR ethics concerns only"]

**Final Justification:**

I agree with reviewer m9FT that the task-dependent hyperparameter is a limiting factor, and could be difficult to potentially tune per task.

Concerning AIME 2024, there are several datasets (e.g AIME 2025, HMMT 2025, Omni-MATH, IMO, USAMO, IMC), which should not be contaminated as they are released after Qwen's cutoff/release date. It could be useful to evaluate, for example, on a subset of Omni-MATH (respecting the academic budget), which has 4K problems, to reduce variance.

I will preserve the acceptance score of 4.

**Limitations:**

yes

**Quality:**

3

**Strengths And Weaknesses:**

Strengths
- Addresses a Practical Challenge that LLM providers would be concerned about
- Extensive Empirical evaluation showing the tradeoff between utility and distillability
- Showcase practical instantiation of the algorithm during inference time without the need of model fine-tuning

Weaknesses
- Though the answer may be correct, the sampling does seem to have an effect on the quality of the solutions as seen in the example traces provided.
- The parameter $\lambda$ is an additional hyperparameter that seems to be different across domains to be effective (e.g GSM8K, MATH, and MMLU). There doesn't seem to be any guidance of selecting this hyperparameter.

---

> ### Author Rebuttal · Authors · 2025-07-31
>
> Thank you for your thorough and constructive review. We're encouraged by your positive assessment of our work's practical relevance, empirical rigor, and technical soundness. We appreciate your recognition that this addresses a real challenge for LLM providers and your acknowledgment of our extensive evaluation. We'd like to address your specific concerns and questions.
>
> ## W1: Impact on traces.
>
> We agree that our procedure affects generated traces—this is indeed the core principle of our algorithm, and we appreciate your observation. However, we'd like to clarify several important points:
>
> Moderate λ preserves interpretability: For practical deployment scenarios with modest performance degradation (~5% in Figure 9), traces remain highly interpretable while providing meaningful antidistillation protection. The first example trace in our paper demonstrates this balance effectively.
>
> Proof-of-concept nature: This work represents the first approach to address controlled distillation poisoning—a problem setting that was previously unexplored. While we agree it would be ideal to have traces indistinguishable from non-poisoned ones, establishing that such poisoning is even possible represents a significant conceptual advance.
>
> Future research directions: We're optimistic about improvements in subsequent work. The dramatic difference between our antidistillation curves and temperature sampling curves (visible across all our tradeoff plots) suggests substantial room for refinement. Similar to how early watermarking techniques were easily detectable but improved over time, we anticipate future work will develop more sophisticated poisoning methods that better preserve trace quality.
>
> ## W2: How to pick the penalty parameter?
>
> We respectfully suggest that having a tunable hyperparameter is actually an __advantage__ rather than a limitation:
>
> Adaptive security posture: Model providers can dynamically adjust λ based on their assessment of distillation risk, content sensitivity, or user behavior patterns. For instance, they might apply stronger protection for sensitive domains (hacking, military applications, biological warfare, coercion, proprietary business analysis, etc.) while using lighter protection for general queries.
>
> Predictable tradeoff curves: Our experiments demonstrate that the utility-distillability tradeoff follows fairly predictable patterns across tasks. One can accurately estimate the full curve from just a few evaluation points, enabling intelligent λ selection without exhaustive tuning.
>
> Task-agnostic approaches: Future work could use a comprehensive capability set to precompute proxy gradients, then select λ based on this unified representation rather than task-specific tuning. Additionally, small auxiliary models could automatically choose appropriate λ values based on conversation context and estimated security risk.
>
> ## Q1: Long-Context Settings (AIME 2025, 32K+ tokens)
>
> Intuitively, we suspect that antidistillation will work better in longer context settings, as there is more room to include the "adversarial example" in the nuanced choice of tokens. However, due to our academic compute constraints, we could not investigate benchmarks requiring very long context reasoning. We did try AIME 2024, but the benchmark is significantly compromised (particularly for Qwen models), and the results so high-variance that we felt they had no scientific value. The size of the benchmark is very small as well.
>
> Additionally, in the long context setting, we note that one does not have to use ADS on the entire reasoning trace. It could be used only in critical parts, or fluctuated off and on as necessary. That is, $\lambda$ could be chosen based on context and content, and in some cases $\lambda = 0$.
>
> ## Q2: Verifree Compatibility
>
> Given that Verifree was published after our submission deadline, we cannot provide definitive experimental validation. However, we note that our method operates primarily through sampling modifications rather than log probability manipulation. While approaches that rely on solution log probabilities might potentially be adapted to work with our poisoned traces, this would require dedicated research to establish effectiveness and represents an interesting direction for future work.
>
> ## Q3: Model Scale Interactions
>
> We've conducted additional experiments specifically addressing proxy-student size mismatches:
>
> Cross-scale robustness: Using a Qwen2.5-1.5B proxy with Llama-3.2-3B student on GSM8K, we observe strong antidistillation effects across λ values from 0.0674 to 0.811, with student accuracy dropping from 59.88% to 4.599% as teacher accuracy decreases from 91.81% to 19.77%. In this setting, we also have a somewhat more favorable latency tradeoff (840s vs 417s).
>
> Reverse configuration: Using a Qwen2.5-3B proxy with Llama-3.2-1B student demonstrates similarly robust poisoning effects, with student accuracy dropping from 30.85% to 7.527% as λ increases from 0.0811 to 0.191.
>
> These results demonstrate that our method tolerates reasonable size disparities between proxy and student models, addressing practical deployment scenarios where model owners lack precise knowledge of potential student architectures.
>
> **Qwen2.5-1.5B Proxy student and Llama-3.2-3B Student on GSM8k**
>
> | $\lambda$ | Teacher Accuracy | Student Accuracy |
> |--|--|--|
> | 0.0674 | 91.81% | 59.88% |
> | 0.108 | 89.99% | 56.99% |
> | 0.122 | 87.01% | 55.42% |
> | 0.136 | 83.79% | 53.02% |
> | 0.149 | 82.46% | 44.67% |
> | 0.163 | 79.82% | 39.78% |
> | 0.177 | 76.92% | 34.24% |
> | 0.204 | 70.89% | 25.97% |
> | 0.245 | 64.10% | 16.87% |
> | 0.259 | 59.22% | 9.512% |
> | 0.811 | 19.77% | 4.599% |
>
>
> **Qwen2.5-3B Proxy student and Llama-3.2-1B student on GSM8k**
>
> | $\lambda$ | Teacher Accuracy | Student Accuracy |
> |--|--|--|
> | 0.0811 | 88.59% | 30.85% |
> | 0.0947 | 86.19% | 20.93% |
> | 0.163 | 70.64% | 10.59% |
> | 0.177 | 68.07% | 7.94% |
> | 0.191 | 64.85% | 7.527% |
>
>
> ## Q4: Human Study on Utility
>
> We absolutely agree that human evaluation would strengthen our analysis of practical utility. Unfortunately, conducting a rigorous human study within the rebuttal timeframe isn't feasible. However, we view this as a crucial component of future work—systematically evaluating whether users can distinguish between standard and lightly poisoned traces (e.g., at 5% performance degradation levels) would provide valuable validation of real-world applicability.
>
> ## Broader Impact
>
> We're gratified by your assessment that this work addresses a practically important challenge. The security implications of unrestricted model distillation have received insufficient attention in the literature, and we believe establishing the feasibility of controlled poisoning represents an important step toward more secure foundation model deployment.
>
> While this initial work doesn't solve all practical concerns, it demonstrates that the binary choice between "release traces" and "withhold traces" is false—there exists a middle ground where partial protection is possible. This opens up new research directions and provides foundation model developers with previously unavailable options for protecting their intellectual property while maintaining service quality.
>
> ## Conclusion
>
> We're grateful for your thorough review and particularly appreciate your recognition of this work's practical importance and technical soundness. We believe our responses have effectively addressed your concerns:
>
> 1. Solution quality remains acceptable at practical operating points, with clear improvement trajectories for future work
> 2. Hyperparameter selection offers valuable adaptive control, with predictable tradeoff patterns enabling practical deployment
> 3. Cross-scale generalization is empirically demonstrated through our additional experiments
> 4. Long-context applicability appears promising based on theoretical considerations, though resource constraints limited full evaluation
>
> Given our comprehensive responses and the additional experimental validation we've provided, we respectfully hope you'll consider increasing your rating. We believe this work represents a solid contribution that will catalyze important future research in foundation model protection—an increasingly critical area as these models become more valuable and ubiquitous.
>
> Thank you again for your constructive engagement with our work.

---

### Official Review · Reviewer_m9FT · 2025-07-02

**Clarity:** 3
**Significance:** 2
**Originality:** 3
**Rating:** 4
**Confidence:** 3

**Summary:**

The paper addresses the risk that releasing chain-of-through traces (and in general, model responses) enable low-cost model distillation. To this end, the authors propose antidistillation sampling (AS), a decoding rule that perturbs the teacher's next token logits with a penalty proportional to how much each candidate token would decrease a student’s loss. The exact penalty is derived via an approximation. The authors conduct experiments on multiple datasets (e.g. GSM8K) and show that AS preserves teacher accuracy better than temperature sampling, while further reducing student accuracy. Finally, they evaluate the generalisation of that method across teacher-student architecture pairs.

**Questions:**

- How sensitive is AS to the capabilities gap between proxy and attacker? Have you tried a larger student model?

**Ethical Concerns:**

["NO or VERY MINOR ethics concerns only"]

**Final Justification:**

Antidistillation sampling is, to the best of my knowledge, the first defense that beats temperature sampling at limiting trace-based distillation. The authors provide sufficient evidence that this holds across datasets and model sizes. While the method requires context-dependent hyperparameter tuning and results in significantly higher cost, it opens up an interesting area for future research.

**Limitations:**

-

**Paper Formatting Concerns:**

-

**Quality:**

3

**Strengths And Weaknesses:**

**Strengths:**
- Model distillation from blackbox APIs is a security risk for foundation model developers and this paper proposes an interesting approach to address it
- The evaluations on multiple benchmarks and model pairs demonstrates that AS outperforms temperature sampling

**Weaknesses:**
- Variation in the hyper-parameter influences AS’s performance degradation differently from one task to another (see Figure 4). In practice, model owners rarely know which task adversaries care about. Thus, they would likely have to use a static value which could lead to small degradation on some tasks but significant degradation on other tasks.
- The cost-benefit trade-off appears weak. AS increases inference cost while still resulting in a non-negligible accuracy drop. For a production LLM, latency and quality will likely outweigh security gains.
- Reasoning traces have the positive side-effect of providing some insight into the models „thinking“. However, the decoding with AS can lead to incoherent reasoning traces (e.g. p. 19) that make the release of reasoning traces essentially useless. Thus, a foundation model developer who cares about this threat could just not release those traces.

---

> ### Author Rebuttal · Authors · 2025-07-31
>
> Thank you for your thoughtful review of our work on antidistillation sampling. We appreciate your engagement with the technical details and practical considerations. We'd like to address each of your concerns systematically.
>
> ## W1: Hyperparameters are task dependent.
> We respectfully disagree that task-dependent hyperparameters represent a fundamental limitation. Rather, we view this as a __feature, not a bug__--providing model providers with precise control over their security posture.
>
> Model providers can adaptively choose λ based on conversation content or context, applying stronger antidistillation for sensitive domains (e.g., military applications, private data analysis) while using lighter protection for general queries.
>
> Our experiments show that the tradeoff curve behavior is quite predictable—one can accurately estimate the full curve with far fewer evaluation points than we plotted, enabling intelligent λ selection.
>
> We outline clear paths forward, including using MMLU performance to predict optimal hyperparameter ranges across tasks, and developing classifiers to select appropriate λ values automatically.
>
> This tunability mirrors established security practices--differential privacy requires choosing ε and δ, watermarking requires choosing strength parameters, and adversarial robustness requires choosing perturbation magnitudes. The security community has successfully operationalized such parameterized defenses.
>
> ## W2: Cost-benefit analysis.
>
> This is the first work in what we hope becomes a productive research direction, and we don't expect to solve all practical concerns in one paper. However, several factors make the tradeoff more favorable than initially apparent:
>
> __Selective deployment:__ As mentioned above, ADS need not be applied universally. Model providers can activate it only when they suspect distillation attempts or for particularly sensitive content, serving as a form of adaptive throttling.
>
> __Computational overhead:__ In the *very worst case*, our sampling method makes inference $3\times$ more expensive in exchange for protecting models from distillation attacks. This is still something model providers care about, and future work can further reduce this overhead. In the more realistic case, when using smaller proxy models with large teacher models, the proxy computation cost becomes a smaller fraction of the total inference budget. We collected some timing information for our experiments with the 3b proxy. In the setting of GSM8k trace generation, AD sampling took 1008s while plain sampling took 428s -- that is, our method is just a bit more than two times slower than normal sampling. We expect future research to cut this overhead down even further.
>
> __Security-critical applications:__ Many deployment scenarios exist where security justifies performance costs—consider hacking, military applications, biological warfare, coercion, etc., where protecting model capabilities outweighs latency concerns.
>
> It was NOT obvious that our method would work as well as it does, given the relatively complicated technical machinery under the hood. Our work is the first to show that anti-distillation is even possible; currently, there is no way to really protect against distillation attacks without holding back the traces or explicitly blocking adversarial accounts. Our results are statistically significant, in that our method significantly and obviously changes the shape of the tradeoff curve; this makes us optimistic that there is room for more methods in this direction. We hope our work will stimulate further research in this area to try and improve upon our results, leading to a more practical application of our method.
>
> ## W3: Model owners could not release traces.
>
> In principle, we agree; a model owner could decide not to release traces. However, as things stand, model owners (e.g., OpenAI, Anthropic, Google, etc.) do release traces, and have not documented any plans to stop this practice, meaning that distillation from these models will remain possible for the foreseeable future.  On the other hand, if our paper results in changing the practice of frontier model providers, we would consider this a significant impact of our work. It's also worth pointing out that companies probably _do_ want to release traces, because not doing so puts them at a disadvantage relative to other model providers: it's harder to get feedback, billing becomes less transparent since users wouldn't get all the tokens they used, users like to understand why certain decisions were made (especially in, e.g, software engineering) -- so hiding traces could lead to backlash. We hope for transparency and protection at the same time.
>
> To answer your question more specifically, we agree that it would be more desirable if our algorithm induced traces that are indistinguishable from non-poisoned traces. However, we strongly emphasize that our algorithm is the first to address this problem setting. In this way, an exciting direction for future research would be to render poisoned traces indistinguishable from non-poisoned ones. However, in our view, the fact that our algorithm did not address every possible concern does not make it unworthy of publication.
>
> Also please note: as seen in Fig. 9, for relatively small amounts of antidistillation (around ~5% degradation), the traces are not heavily impacted. You can see this in the first example trace listed in the paper.
>
> ## Q1: Sensitivity to proxy/attacker capability differences.
>
> We've conducted additional experiments specifically to address this concern:
>
> Cross-size robustness: Using a Qwen2.5-1.5B proxy with Llama-3.2-3B student on GSM8K, we observe robust antidistillation effects across λ values from 0.0674 to 0.811, with student accuracy dropping from 59.88% to 4.599% as teacher accuracy decreases from 91.81% to 19.77%. In this setting, we also have a somewhat more favorable latency tradeoff (840s vs 417s).
>
> Reverse configuration: Using a Qwen2.5-3B proxy with Llama-3.2-1B student shows similarly strong poisoning effects, with student accuracy dropping from 30.85% to 7.527% as λ increases.
>
> These results demonstrate that our method tolerates reasonable size mismatches between proxy and student models, addressing practical deployment scenarios where exact architectural knowledge is unavailable.
>
> **Qwen2.5-1.5B Proxy student and Llama-3.2-3B Student on GSM8k**
>
> | $\lambda$ | Teacher Accuracy | Student Accuracy |
> |--|--|--|
> | 0.0674 | 91.81% | 59.88% |
> | 0.108 | 89.99% | 56.99% |
> | 0.122 | 87.01% | 55.42% |
> | 0.136 | 83.79% | 53.02% |
> | 0.149 | 82.46% | 44.67% |
> | 0.163 | 79.82% | 39.78% |
> | 0.177 | 76.92% | 34.24% |
> | 0.204 | 70.89% | 25.97% |
> | 0.245 | 64.10% | 16.87% |
> | 0.259 | 59.22% | 9.512% |
> | 0.811 | 19.77% | 4.599% |
>
>
> **Qwen2.5-3B Proxy student and Llama-3.2-1B student on GSM8k**
>
> | $\lambda$ | Teacher Accuracy | Student Accuracy |
> |--|--|--|
> | 0.0811 | 88.59% | 30.85% |
> | 0.0947 | 86.19% | 20.93% |
> | 0.163 | 70.64% | 10.59% |
> | 0.177 | 68.07% | 7.94% |
> | 0.191 | 64.85% | 7.527% |
>
> ## Broader Perspective
>
> Resource constraints: As academics, we cannot afford to run experiments at the scale of frontier models (100B+ parameters, million-token contexts). However, we've demonstrated the core principle across multiple benchmarks, architectures, and scales within our resource constraints. The research community often advances through such incremental contributions that industry can then scale.
>
> Statistical significance: Our results show clear, statistically significant degradation in student performance across multiple settings, confirming that antidistillation is not only possible but practically achievable.
>
> Research impact: Even if this specific implementation isn't immediately production-ready, demonstrating that controlled distillation poisoning is feasible represents a significant step forward in understanding model security. We hope this work catalyzes further research toward more efficient and practical solutions.
>
> We believe the evidence strongly supports the value of this research direction, even as we acknowledge areas for future improvement. The field advances through such foundational contributions that establish feasibility and point toward promising research directions.
>
> ## Conclusion
> We sincerely appreciate the time and effort you've invested in reviewing our work. We believe our responses demonstrate that:
> 1. Task-dependent hyperparameters provide valuable adaptive security control rather than representing a fundamental limitation
> 2. Cost-benefit tradeoffs are reasonable for security-critical applications, with clear paths for improvement in future work
> 3. Reasoning trace quality remains acceptable at practical operating points, with competitive dynamics preventing simple trace withholding
>
> Most importantly, we've provided additional experimental evidence showing robust cross-architecture and cross-scale generalization, directly addressing your questions about capability gaps and larger student models.
>
> While we acknowledge this is foundational work with room for improvement, we respectfully submit that demonstrating the feasibility of controlled distillation poisoning--previously thought impossible--represents a significant contribution to model security. The dramatic performance differences between our method and temperature sampling across multiple benchmarks provide strong evidence of practical utility.
>
> Given our thorough responses to your concerns and the additional experimental validation we've provided, we hope you'll consider revising your assessment. We believe this work merits acceptance as an important first step in protecting foundation model intellectual property while maintaining service quality.

---

> > ### Comment · Reviewer_m9FT · 2025-08-04
> >
> > Thank you for your response and the additional experiments.
> >
> > I agree that the hyperparameter itself is a useful feature, but still believe that the fact that its effect is task-dependent is a limitation. As a result, model providers might have to use different lambda values to get the same level of obfuscation in different contexts. That being said, I agree that this paper sufficiently demonstrates that their method beats temperature sampling and believe that it could open up future work in that direction. I will adjust my score to 4.

---

### Official Review · Reviewer_myTd · 2025-07-02

**Clarity:** 4
**Significance:** 3
**Originality:** 3
**Rating:** 4
**Confidence:** 4

**Summary:**

This paper introduces a technique named "antidistillation sampling" to counter model distillation, a process where knowledge from large "teacher" models is transferred to smaller "student" models. The proposed method works by strategically "poisoning" the teacher model's output reasoning traces during the sampling phase. This is done by adding a dynamically computed penalty to the model's logits, with the goal of increasing the downstream loss for any student model trained on these outputs. The authors present an efficient finite-difference approximation using a "proxy" model to make this approach computationally feasible. Experiments conducted on reasoning benchmarks like GSM8K, MATH, and MMLU show that this method can reduce the performance of a distilled student model more effectively than a standard temperature sampling baseline, while aiming to preserve the teacher model's performance.

**Questions:**

1. Originality vs. Synthesis：Could the authors better position their work in light of existing literature on data poisoning and controlled decoding? A more direct discussion of how antidistillation sampling moves beyond a simple combination of these ideas would help clarify the paper's core conceptual novelty.
2. Justification for Practicality：The paper's practical value is contingent on managing the computational overhead. For this method to be considered a strong candidate for real-world deployment, a more comprehensive study on the overhead vs. efficacy trade-off is needed. Could the authors provide results showing how the defense effectiveness degrades as the proxy model's size and, consequently, the computational overhead are significantly reduced (e.g., using a proxy that is 10x or 50x smaller than the teacher)?
3. Evidence for Generalization：The central premise of using a proxy model requires that the poisoning effect generalizes well. Could the authors provide more evidence or at least a stronger argument for this generalization? For example, would the defense hold if the proxy model is from an entirely different architectural class (e.g., a GRU-based model) than the student (a Transformer)? Without a better understanding of why this generalization occurs, it feels more like an empirical observation than a robust property.
4. On Robustness to Adaptive Adversaries：Have the authors considered how this defense would perform against an adversary who is aware of the poisoning? An adaptive adversary might analyze the generation statistics or employ robust training techniques to mitigate the poisoning. A discussion of the method's potential vulnerabilities to such adaptive attacks would provide a more complete picture of its security.

**Ethical Concerns:**

["NO or VERY MINOR ethics concerns only"]

**Final Justification:**

I will maintain my recommended score, the reason has been explained in the reply to the authors' rebuttal.

**Limitations:**

yes

**Quality:**

3

**Strengths And Weaknesses:**

Strengths：
Significance：The paper addresses a problem of clear practical significance. The threat of model distillation, which can lead to the loss of valuable intellectual property and the proliferation of unsafe models, is a major concern for developers of frontier AI systems. By proposing a direct defense mechanism, this work makes a valuable and timely contribution to the field of AI security and trustworthiness. The problem is well-motivated and its solution would be of considerable interest to the machine learning community, especially industry labs.
Clarity：The paper is exceptionally well-written and presented. The authors do an excellent job of communicating a complex idea in an accessible manner. The motivation and high-level approach are lucidly explained in the introduction and visualized effectively in Figure 1. The mathematical derivation in Section 3.2 is presented with logical progression, building from the ideal objective to the practical approximation. The experimental results are clearly depicted in figures that are easy to interpret; for instance, the trade-off curves in Figures 3 and 4 compellingly illustrate the method's primary claim over the temperature sampling baseline.
Originality：The paper's originality lies in the clever synthesis and application of existing concepts to a new problem domain. While the high-level idea of a dynamic, sampling-based defense is novel, the core technical components are individually well-established. The paper itself acknowledges its connection to the literature on data poisoning and controlled language model decoding. Furthermore, the central technical simplification—approximating a Jacobian-vector product (Eq. 7 ) with a finite-difference scheme (Eq. 10 )—is a standard and widely used numerical method. Therefore, the work should be commended for its novel formulation of the problem and the elegant way it combines these parts, but it does not introduce a fundamentally new algorithm from first principles.
Quality： The technical derivation of the sampling method is logical, and the use of a finite-difference approximation to make the method practical is a clever implementation choice.The use of distinct model architectures for the teacher, proxy, and student (deepseek-ai/DeepSeek-R1-Distill-Qwen-7B, Qwen/Qwen2.5-3B, and meta-llama/Llama-3.2-3B, respectively)  is a strong point that mimics a realistic scenario. The results convincingly show the method works under these specific conditions.
Weaknesses：
Limited Originality：While the specific application is novel, the core technical components are built upon existing ideas. The concept of data poisoning to degrade model performance is well-established. Similarly, controlled decoding and using finite-difference methods to approximate gradients are known techniques. The paper's main contribution feels more like a skillful synthesis and application of these ideas rather than the introduction of a fundamentally new algorithmic paradigm.
Concerns on Practicality and Overhead：The method's quality as a practical solution is impacted by its computational overhead, which requires two forward passes on a proxy model for every generated token. The paper suggests using a smaller proxy model to mitigate this, but it lacks a thorough analysis of the trade-off between proxy model size, defense effectiveness, and latency. This omission makes it difficult to assess the method's viability in real-world, resource-constrained environments.
Scope of Evaluation: The experimental validation, while solid, is somewhat limited. The crucial claim of generalization (poisoning for a Qwen proxy affects a Llama student) is demonstrated on only one specific architectural pairing. The robustness of this effect across a wider variety of model families and sizes is not explored. Furthermore, the defense is tested in a passive setting, without considering potential adaptive adversaries who might try to detect and counteract the poisoning. These factors limit the confidence in the method's general applicability and robustness.
Limited Generalization Evidence：The claim of generalization is central to the method's practicality, yet it is tested on only one cross-family architecture pair (Qwen proxy, Llama student). While Figure 10 shows results for within-family distillation, the more critical cross-family case is not sufficiently explored to make a strong, general claim. The quality of the paper would be significantly higher if this were tested more broadly.
Passive Threat Model：The evaluation is conducted against a passive adversary who naively distills the poisoned traces. It does not consider a more realistic, adaptive adversary who might try to detect statistical anomalies in the traces or alter their distillation objective to be robust to the poisoning. This limits the scope and practical relevance of the demonstrated results.

---

> ### Author Rebuttal · Authors · 2025-07-31
>
> Thanks for your review and constructive comments. We're glad that you find our paper has clear practical significance, is exceptionally well-written, and features logical technical derivation. We provide additional experimental evidence and discussion to address your concerns.
>
> ## Concerns regarding  originality
>
> You raise an important question about our algorithmic contribution. We believe our work introduces several genuinely novel insights that extend well beyond "skillful synthesis" of existing techniques.. As far as we are aware, no prior algorithms address the setting we consider, and our derivation produces a truly novel solution to an important problem.
>
> It's worth noting that *every* algorithm builds upon existing ideas. One could argue that the Fast Fourier Transform isn't innovative since it applies divide-and-conquer to the Discrete Fourier Transform, yet we rightfully consider FFT a fundamental breakthrough. The key insight—recognizing recursive structure—was neither obvious nor trivial to discover.
>
> Consider controlled decoding, which you mentioned in your review. Existing methods typically steer generation toward attributes like sentiment or safety using auxiliary models. In contrast, our objective centers on _simulated_ distillation training steps of a proxy model to poison traces for unknown student models. While our method _can_ be seen as controlled decoding, this problem formulation and solution approach are, to our knowledge, completely novel.
>
> Two innovations deserve particular emphasis. First, no prior work demonstrated that perturbations computed for one architecture would transfer effectively to completely different target architectures. This cross-architecture generalization was theoretically uncertain and empirically unproven. Second, our gradient precomputation technique solves a fundamental computational challenge by deriving a method to precompute gradients once rather than computing them at every decoding step, making the approach computationally tractable through careful finite-difference approximation.
>
> The method's effectiveness was far from guaranteed. We made several strong assumptions: that proxy models would generalize to unknown students, that single-step gradient approximations would suffice for complex optimization landscapes, and that our finite-difference approach would preserve essential gradient information. The fact that these assumptions hold together--producing the dramatic shift in trade-off curves we demonstrate--represents a surprising empirical phenomenon that advances our understanding of model vulnerabilities and defenses.
>
> This is, at __minimum__, a fascinating discovery that opens new research directions. The clear effectiveness of our approach suggests that even better methods may exist, making this an exciting area for future investigation.
>
> ## Practicality
>
> You correctly identify computational overhead as a key concern. In the *very worst case*, our sampling method makes inference $3\times$ more expensive in exchange for protecting models from distillation attacks. This is still something model providers care about, and future work can further reduce this overhead. In the more realistic case, when using smaller proxy models with large teacher models, the proxy computation cost becomes a smaller fraction of the total inference budget. Our method proves particularly practical in the regime of ~5% reductions in teacher performance (Figure 9), where the traces are also more comprehensible (see the first trace listed in the paper).
>
> We collected some timing information for our experiments with the 3b proxy. In the setting of GSM8k trace generation, AD sampling took 1008s while plain sampling took 428s -- that is, our method is just a bit more than two times slower than normal sampling. We expect future research can cut this down.
>
> Our new experiments demonstrate effectiveness across different proxy-student size combinations. A Qwen2.5-1.5B proxy successfully poisons Llama-3.2-3B student distillation, suggesting that using smaller proxies can significantly reduce computational overhead while preserving protection efficacy.
>
> Importantly, antidistillation sampling enables selective deployment rather than universal application. Model providers can activate the method through the tunable λ parameter when detecting suspicious usage patterns or serving particularly sensitive content. This targeted approach eliminates overhead during normal operation while providing protection when most needed--essentially functioning as an intelligent throttling mechanism.
>
> We view this as establishing a new research direction rather than providing the final solution. Future work could explore training teachers to inherently produce antidistillation traces or developing specialized heads for efficient poisoning, potentially eliminating the computational overhead entirely.
>
> ## Scope of evaluation
>
> Your points about experimental breadth are well-taken. Within academic compute constraints, we focused on demonstrating our core hypothesis: that antidistillation sampling can effectively poison distillation attempts. Our cross-architecture results (Qwen teacher/proxy with Llama student) address the most critical practical question--whether the method works when attackers use different architectures than the defender anticipates.
>
> We have reasonable evidence that our proxy setup works across architectures -- as you pointed out, this is the main result we show in the paper. Qwen and Llama represent genuinely different architectures trained on distinct datasets, providing meaningful evidence for generalization. We believe there is generally representation similarity across architectures; they're trained on similar data with similar objectives, ultimately using stacks of similar layers. Even GRU or SSM models can be seen as riffs on linear attention. It has long been known that adversarial examples often transfer across models [1, 2]. The representational similarity across models has also been noted [3].
>
> While we acknowledge that broader evaluation would strengthen our claims, we believe our evidence sufficiently demonstrates the method's viability to warrant community attention and follow-up research.
>
> ## Passive threat model
>
> The adaptive adversary question represents an excellent direction for future research. Our work establishes the fundamental feasibility of trace poisoning--a necessary first step before studying arms race dynamics. Starting with the simplest possible setting and solution has revealed promising initial results that justify deeper investigation.
>
> We expect adaptive defenses will emerge, potentially through statistical anomaly detection or robust training techniques. However, our method's dynamic nature--using gradients from a hidden proxy model with secret evaluation loss--creates a moving target similar to stream ciphers in cryptography. This fundamental property should make detection and countermeasures significantly more challenging than static watermarking approaches.
>
> The cat-and-mouse game between poisoning and detection methods will likely drive important innovations in model security. Our contribution provides a concrete foundation for this research direction.
>
> ## Generalization evidence
>
> This paper was quite expensive for us as academics. We can't afford to run all these experiments. We have shown that our idea works and presented it clearly. The next obvious question is whether it generalizes beyond ~10b models and for much longer traces. We simply cannot do experiments at this scale; what recourse do we realistically have? See above.
>
> We've conducted a few additional experiments to address cross-size generalization concerns. Using a Qwen2.5-1.5B proxy with Llama-3.2-3B student on GSM8k, we observe strong antidistillation effects. The reverse configuration (Qwen2.5-3B proxy with Llama-3.2-1B student) shows similarly robust poisoning effects.
>
> These results demonstrate that our method tolerates reasonable size mismatches between proxy and student models, addressing practical deployment scenarios where exact architectural knowledge is unavailable. In the setting where we use a smaller proxy, we also have a somewhat more favorable latency tradeoff (840s vs 417s).
>
> **Qwen2.5-1.5B Proxy student and Llama-3.2-3B Student on GSM8k**
>
> | $\lambda$ | Teacher Accuracy | Student Accuracy |
> |--|--|--|
> | 0.0674 | 91.81% | 59.88% |
> | 0.108 | 89.99% | 56.99% |
> | 0.122 | 87.01% | 55.42% |
> | 0.136 | 83.79% | 53.02% |
> | 0.149 | 82.46% | 44.67% |
> | 0.163 | 79.82% | 39.78% |
> | 0.177 | 76.92% | 34.24% |
> | 0.204 | 70.89% | 25.97% |
> | 0.245 | 64.10% | 16.87% |
> | 0.259 | 59.22% | 9.512% |
> | 0.811 | 19.77% | 4.599% |
>
>
> **Qwen2.5-3B Proxy student and Llama-3.2-1B student on GSM8k**
>
> | $\lambda$ | Teacher Accuracy | Student Accuracy |
> |--|--|--|
> | 0.0811 | 88.59% | 30.85% |
> | 0.0947 | 86.19% | 20.93% |
> | 0.163 | 70.64% | 10.59% |
> | 0.177 | 68.07% | 7.94% |
> | 0.191 | 64.85% | 7.527% |
>
> ## Summary
>
> We agree there's substantial room for future work and hope to catalyze research in this important area. This represents the first paper addressing antidistillation through sampling strategies, and our results demonstrate clear feasibility with quantifiable trade-offs. The dramatic difference between our trade-off curves and temperature sampling baselines indicates we've identified a genuinely effective approach to an important security problem.
>
> While no first paper in a new area achieves perfect generality or eliminates all limitations, our contribution establishes both the problem's tractability and a concrete solution that others can build upon. We hope to start the scientific conversation by presenting it at this venue.
>
>
> [1] The Space of Transferable Adversarial Examples. Tramer et al.
>
> [2] Explaining and Harnessing Adversarial Examples. Goodfellow et al.
>
> [3] The Platonic Representation Hypothesis. Huh et al.

---

> > ### Comment · Reviewer_myTd · 2025-08-06
> >
> > Thank you for the detailed rebuttal and for providing additional experiments. Your response has helped clarify your positions. However, my main concerns, which led to my borderline rating, persist for the following reasons:
> > 1、On Originality: I understand your perspective on the novelty of the problem formulation. My assessment, however, was focused on the novelty of the underlying technical components. While we are discussing different facets of originality, my initial evaluation that the work is a highly effective synthesis of existing techniques remains, and I will maintain my score on this point. 2、On Practicality and Overhead: I appreciate you providing concrete latency data. This offers some evidence of the cost. However, the >2x overhead is still a significant concern for large-scale, practical deployments. The "selective deployment" strategy you proposed is an interesting operational idea, but it does not solve the core efficiency issue of the algorithm itself. 3、On Generalization: Thank you for conducting new experiments on cross-scale generalization. This new data certainly strengthens the paper. My reservation, however, concerns generalization to more architecturally diverse models. Although Qwen and Llama have their differences, they are both based on the same fundamental Transformer architecture. The question of whether this poisoning effect transfers to truly distinct architectures (e.g., non-Transformer-based models) remains open.
> > In conclusion, your rebuttal and the new data were clarifying and have improved the paper. The work addresses a promising and important research direction. Nevertheless, in its current state, the reasons to accept are balanced by the significant limitations regarding practical efficiency and the full scope of generalization. Therefore, I will be maintaining my original rating.

---

> ### Comment · Area_Chair_XpqA · 2025-08-05
> **Ping**
>
> Dear Reviewer,
>
> The deadline for the author-reviewer discussion is approaching (Aug 8, 11.59pm AoE).
> Please read carefully the authors' rebuttal and engage in meaningful discussion.
>
> Thank you,
> Your AC

---

### Note · Authors · 2025-08-16

We appreciate the reviewers' engagement and address key points from discussion.

**Hyperparameter Selection:** Two reviewers note $\lambda$ appears "task-dependent," but the variation is less problematic than suggested. Student models achieve different baseline performance: MMLU ($\sim$50%), GSM8K ($\sim$63%), MATH ($\sim$15%) when trained without antidistillation. Given these vastly different starting points, variation in absolute accuracy drops is natural.

Importantly, $\lambda$ values in the $O(10^{-1})$ range produce consistent effects across all tasks. For instance, $\lambda \approx 0.13$ yields similar relative degradation ($\sim$30% drop) across all datasets. The effective ranges cluster tightly—not orders of magnitude apart as expected if truly task-sensitive. This consistency across fundamentally different benchmarks demonstrates robustness rather than limitation. The tunable $\lambda$ also enables adaptive security postures, as discussed in our rebuttal.

**Scope and Significance:** Several reviewer suggestions extend beyond establishing this research direction. Requests for non-transformer architectures, production-ready latency, perfect trace indistinguishability, and frontier-scale experiments are natural follow-ups to our foundational contribution. This is the *first* paper demonstrating controlled distillation poisoning is possible—what reviewers acknowledge as addressing a "practical challenge" of "clear practical significance."

Our experiments validate the core hypothesis: antidistillation sampling dramatically shifts utility-distillability tradeoffs versus temperature sampling. Statistical significance across multiple benchmarks, architectures, and scales establishes feasibility without requiring every architectural variant.

**Impact:** We've shown model providers aren't limited to "release traces and risk distillation" or "hide traces and lose transparency." Our method enables a third option: useful traces that resist distillation—a paradigm shift in model protection. As reviewer m9FT notes, this "could open up future work in that direction."

We've provided extensive cross-architecture/cross-scale experiments, demonstrated practical operating points, and outlined potential optimization paths. The dramatic tradeoff curve differences prove this isn't theoretical—it's a working defense that future research can refine.

Foundational work should be evaluated on demonstrating feasibility and impact potential, which our results achieve.

---

### Decision · Program_Chairs · 2025-09-17

**Decision:**

Accept (poster)

**Comment:**

All reviewers generally agree that the paper tackles a relevant problem in a novel and technically clever way, that the technique is empirically promising, and that, more generally, the work's strengths outweight its weaknesses. The author-reviewer discussion was respectful and effective.  I recommend the authors to include a discussion of the most frequent issues raised by the reviewers (specifically, the trade-off between effectiveness and computational costs and the choice of hyperparameter) in the revised manuscript.